# Wearable Biosensor Standardization: How to Make Them Smarter

Giada Giorgi * and Sarah Tonello

Department of Information Engineering, University of Padova, 35131 Padua, Italy; sarah.tonello@unipd.it
* Correspondence: giada.giorgi@unipd.it

**Abstract:** The availability of low-cost plug-and-play devices may contribute to the diffusion of methods and technologies for the personalized monitoring of physiological parameters by wearable devices. This paper is focused on biosensors, which represent an interesting enabling technology for the real-time continuous acquisition of biological or chemical analytes of physio-pathological interest, e.g., metabolites, protein biomarkers, and electrolytes in biofluids. Currently available commercial biosensors are usually referred to as customized and proprietary solutions. However, the efficient and robust development of e-health applications based on wearable biosensors can be eased from device interoperability. In this way, even if the different modules belong to different manufacturers, they can be added, upgraded, changed or removed without affecting the whole data acquisition system. A great effort in this direction has already been made by the ISO/IEC/IEEE 21451 standard that introduces the concept of smart sensors by defining the main and essential characteristics that these devices should have. Following the guidelines provided by this standard, here we propose a set of characteristics that should be considered in the development of a smart biosensor and how they could be integrated into the existing standard.

**Keywords:** standards; smart biosensors; wearable devices; transducer electronic data sheet





## 1. Introduction

Biosensors are a particular kind of sensing devices for the measurement of biological or chemical reactions that generate signals proportional to the concentration of a given analyte. In recent years, a large number of applications based on wearable biosensors have been proposed [1]. These applications cover several fields ranging from environmental monitoring to disease detection, food safety, defense, drug discovery, etc. Recent achievements in innovative fabrication technologies (e.g., rapid prototyping and printed electronics) enable integration of various types of biosensors within wearable devices [2–4]. This contributed to the acceleration of their use for the continuous real-time monitoring of physiological parameters and for non-invasive measurement of biochemical markers in biofluids, such as sweat, tears, and saliva [5]. Furthermore, the possibility of integrating biosensors with smartphones or other ubiquitous smart mobile devices, e.g., smartwatches, has driven the development of low-cost applications for pervasive monitoring of the health status of people [6,7]. Examples of such applications include physical exercise performance monitoring [8], health care applications for the control of diabetes and other diseases [9] and home monitoring of the elderly [10].

The development of applications based on wearable biosensors can be greatly facilitated by the availability of smart devices, in particular when a standardized language is adopted to describe and query them. An important requirement for an efficient and, at the same time, robust development of data acquisition systems is the possibility to guarantee device interoperability, even when made by different manufacturers. This feature is referred to as plug-and-play capability, thus referring to very different devices that can be added, upgraded, changed or removed without affecting the whole data acquisition system [11].

The definition of standardized procedures is the first step in the development of plug-and-play wearable biosensors since, in this way, their common features can be described

with a unified language. This step enables adding new biosensors, upgrading existing ones, removing obsolete devices and, finally, integrating biosensors with other kinds of sensors, such as temperature and humidity, in a smart and effective way and also keeping integration costs low for application developers. Furthermore, as claimed in the emerging Internet of Things (IoT) paradigm, wearable biosensors, such as other sensors, need to be addressed and connected to the user network in a unique and secure way.

For sensors commonly employed in industrial, entertainment, environmental and smart home applications, the ISO/IEC/IEEE 21451 standard family provides a set of valuable features, introducing and standardizing the concept of a smart transducer [12]. The logical structure of a smart transducer consists of two logical modules: a *transducer interface module* (TIM) and a *network-capable application processor* (NCAP). The TIM module provides an interface toward the physical sensors and actuators which represent the interface, referred to as a *transducer channel* (TCh) between the physical and information domains. The NCAP supports the communication toward the user network and one or more TIMs, thus acting as a sort of access point between the local transducer network, composed of the TIM nodes and the user network/internet. NCAP and TIM modules are allowed to communicate through a standardized *transducer independent interface* (TII), which can be designed by considering different solutions (WiFi, Bluetooth, Zigbee, etc.) with wireless interfaces standardized by ISO/IEC/IEEE 21451.5 (in short Dot 5). The proposed logical structure for a smart transducer enables separation of aspects strictly related to data communication from those concerning the sensing and acquisition process. The standard ISO/IEC/IEEE 21451.0 (or Dot 0) introduces the definition of transducer electronic data (TEDS) to add self-awareness to a smart transducer. TEDSs are essentially standardized electronic documents that provide a comprehensive description of the characteristics concerning the transducer, the data acquisition system, the logical structure of the smart sensors and communication protocols. Information specified in the TEDS enables complete characterization of the behavior of a given transducer as well as the kind of measurement that can be performed with that device.

The main advantage of the availability of ISO/IEC/IEEE 21451-compliant biosensors is easier integration in vendor-independent applications, where other kinds of sensors can also be employed. This will also reduce production costs, guaranteeing more robustness and, at the same time, enable a more efficient use of the available resources. Furthermore, there is the possibility to integrate TEDS at the sensor as a printed TEDS as described in [13], which consists of a QR code that can directly be decoded by image scanners. This represents a cheap way to disseminate transducer specifications since information about the sensor or biosensor can be directly printed on any surface and stored without any energy consumption.

In this picture, this paper aims to propose a solution for integrating biosensors in the IEEE 21451 standard family. The first step to achieve this purpose consists of comparing and evaluating the differences that a biosensor possesses compared to a 'normal' sensor. Biosensors have important peculiarities, such as the presence of analytes and bioreceptors in their structure, the need for frequent and accurate calibrations and the need to properly initialize the measurement setup before starting to acquire measured values [14]. This information should be stored in the system and a generic user should be able to access the information when required. In general, the calibration curve for a biosensor, different to that of other kinds of sensors, is characterized by poor accuracy. It is preferable to directly provide information obtained for different calibration points instead of an approximating mathematical function; this will provide a greater flexibility for the app designer that can consider the best interpolation method for estimating the calibration curve on the basis of the particular kind of application and input range for the concentration of the substance of interest, as will be explained in the following [15].

This paper is organized as follows. A description of the main characteristics and parameters of a biosensor is first provided in Section 2. After that, in Section 3, a detailed description of what should be referred to as a smart transducer is addressed. Following,

the second step, explained in Section 4, consists of proposing modifications to the TEDSs for describing the main characteristic of a biosensor. Finally, in Section 5, some conclusions about this work are reported.

## 2. ISO/IEC/IEEE 21451.0 Smart Transducer

The wearable system described in our previous work [16] will be considered in the following to introduce the standardized approach for biosensors. This multi-sensing system, illustrated in Figure 1, integrates a superficial multi-electrode electromyographic sensor (sEMG) on the same device composed of eight differential channels and a biosensor for the measurement of the lactate concentration in sweat, which represents a useful aid in applications that aim at estimating muscular fatigue during physical exercise [17]. Signals provided by the physical sensors are acquired by a multi-channel acquisition system, which operates as a parallel acquisition device, composed of eight different analogue-to-digital converters (ADCs) for the simultaneous acquisition of the signals provided by the sEMG sensor. Another ADC is further employed for acquisition of the signal provided by the lactate biosensor (Figure 2). Data acquired from this multi-channel acquisition system are transferred by a serial peripheral interface (SPI) bus to an external microcontroller, where they are elaborated and temporarily stored in dataset to be finally transmitted by a Bluetooth low-energy (BLE) wireless interface to the final user. Without the lack of generality, we can consider this example to introduce the different aspects of the Dot 0 standard.

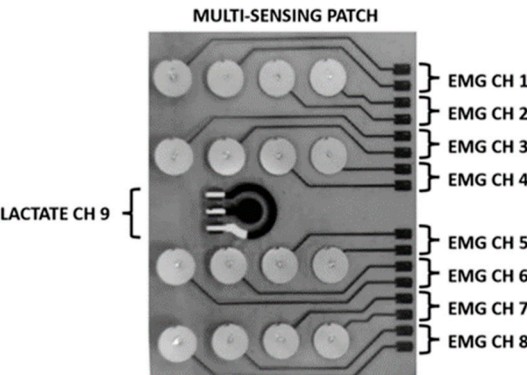

**Figure 1.** Multi-sensing system, described in our previous work [17], that simultaneously measures muscular fatigue by a sEMG array (EMG CH 1–8) and lactate concentration (LACTATE CH 9).

The questions that we wish to answer are: How much and what kind of information should an app developer have to properly elaborate the acquired data? What parameters of this system should an app designer be able to change to guarantee the desired accuracy and reliability for a given measurement application? For instance, the considered system supports acquisition from nine different sensors simultaneously. However, due to network bandwidth constraints, sometimes it could be useful to limit acquisition to only 4 or 5 sensors. This should be possible without changing the hardware setup. As described in Section 4.1, TCHs can be organized by the manufacturer in different proxy structures. This would allow an app designer to easily select the desired group of sensors and finally read data only from the selected sensor subset.

Figures 3 and 4 illustrate a standardized version, compliant with the IEEE 21451 standard of the previous system. Figure 3 describes the system at the network level: the local sensing and acquisition system is represented by the TIM module, which is connected to the gateway NCAP—for instance, a smartphone—through the TII interface that, for wireless protocols such as BLE, is standardized by Dot 5. Therefore, all the BLE protocol parameters, i.e., bandwidth, payload size, timing information, etc., can be easily found in the corresponding Dot 5 TEDS. Standardized procedures are specified in the Dot 5 document, which provides a simple and flexible method to change the setup for

the wireless interface. Figure 4 instead provides the standardized description of the local node, -TIM.

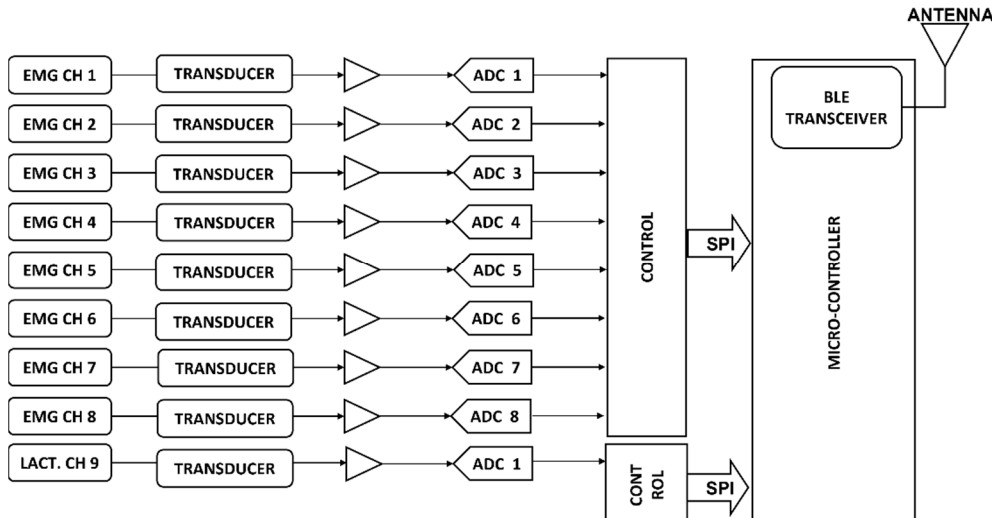

**Figure 2.** Multi-channel acquisition system to interface at the multi-sensing system in Figure 1. EMG = electromyographic sensor; LACT = lactate biosensor.

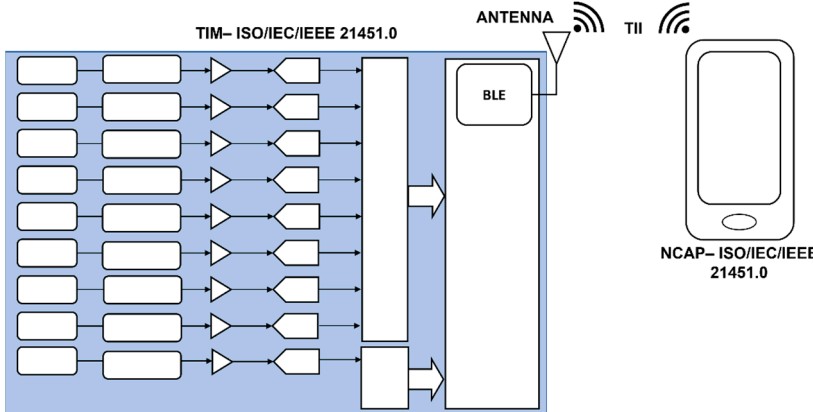

**Figure 3.** Network representation for the wearable sensing system of Figure 2. Block names have been omitted, in order to focus on the main Dot 0 blocks: TIM, TII and NCAP. Anyway, they are represented by using the same symbol of Figure 2.

The transducer channel represents the first block of a smart sensor, as described in the Dot 0 document. This block includes a physical sensor, conditioning electronics (i.e., transducers, analogue filters, amplifiers, etc.) and an ADC converter [18]. A transducer channel must be defined for each physical sensor in the TIM. In the example of Figure 4, there are nine transducer channels, eight for each channel of the sEMG sensor and one for the lactate biosensor. It is very important to observe that TCh organization does not depend on the particular kind of physical sensor. This means that the same standardized procedures—setup commands, read commands, etc.—can be employed to acquire data both from an EMG sensor and from the lactate biosensor. Furthermore, a standardized description of the sensor type and corresponding metrological characteristics is provided for each TCh in the TIM; this is made possible by using the TEDSs. The availability of standardized descriptions and procedures greatly facilitates the development of new applications. Nevertheless, biosensors present some important peculiarities compared to other kind of sensors that are not taken into consideration by the current version of the IEEE 21451; for this reason, this standard should be eventually extended. In this paper, we aim at provide an in-depth analysis of the main parameters that an app designer should consider

when dealing with biosensors. This analysis enables identifying the main problems, for which we propose practical solutions.

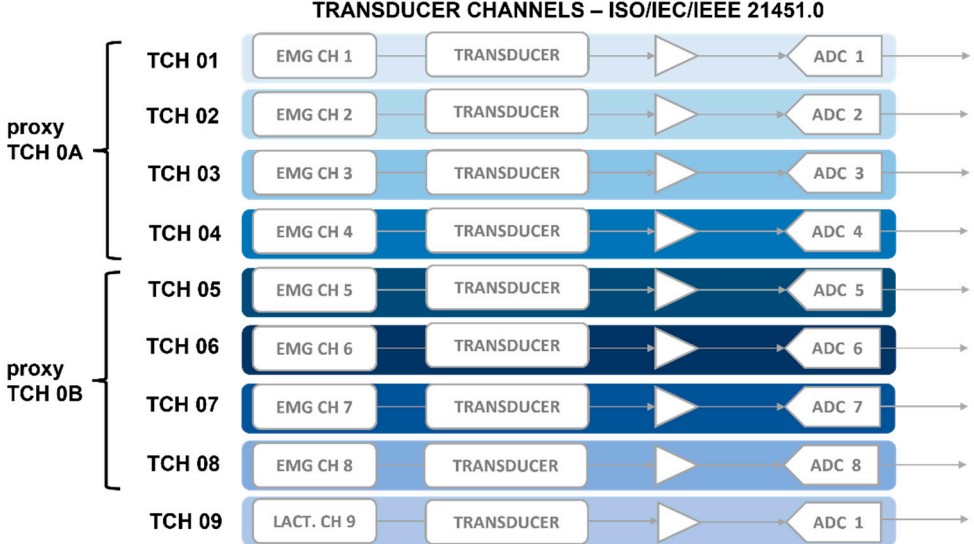

**Figure 4.** Standardized description compliant with the IEEE 21451 family of the local node described in Figure 2.

### 3. Smart Biosensor: A Short Recall

A sensor is a device that, when affected by an input physical quantity, e.g., pressure, heat, humidity, movement, and force, provides an output signal that can be easily acquired and analyzed [14]. Unlike an analog sensor, which provides an electrical signal as output, a smart sensor outputs a digital number that directly represents an input quantity estimate. For this reason, a smart sensor integrates, within the same physical device, the sensing part, the acquisition stage, elaboration, and transmission blocks.

Recently, the availability of new materials and new fabrication technologies has led to the commercialization of wearable biosensors for the continuous monitoring of physiological parameters in real time. Similarly, smart biosensors include processing and transmission blocks, to enable the development of properly stand-alone wearable devices. The structure of a biosensor is, however, slightly different from that of a common sensor and can be synthesized by three blocks—*analyte*, *bioreceptor*, and *transducer* [14]—as illustrated in Figure 5. The analyte is the substance of interest that the biosensor must detect and measure, where $x(t)$ represents the concentration of the analyte in a given time $t$. For instance, in a biosensor designed to detect glucose, the latter represents the analyte and $x(t)$ will represent the glucose concentration. The bioreceptor is a molecule that specifically recognizes the analyte; examples of bioreceptors are enzymes, cells, aptamers, deoxyribonucleic acid (DNA), and antibodies [19].

The process involving the signal generation (e.g., a light, heat, and pH variation) resulting from the interaction of the bioreceptor with the analyte is usually called *bio-recognition*. In a biosensor, the transducer converts the bio-recognition event into a measurable signal. A transducer produces an optical, electrical or piezoelectric signal $v(t)$ that is usually proportional to the number of analyte–bioreceptor interactions [6]; this procedure is also referred to as *signalization*. The output of the transducer is then digitalized by an ADC that provides a flow of binary codewords $y(n)$. These codewords are then converted into a digital signal $v_Q(n)$ that corresponds to a digital approximation of the analog signal $v(t)$ provided in the output by the transducer. Finally, the elaboration block makes use of information related to the calibration curve—for instance, offset $q$ and sensitivity $m$ in a linear device—to calculate a digital representation of the analyte concentration $\hat{x}(n)$. Information on the calibration curve must be stored in the local memory within the smart biosensor. For a generic sensor, calibration curve parameters are directly specified in the Calibration TEDS.

For a biosensor, however, it is more convenient to provide all the information obtained during the calibration procedure, letting the app designer calculate the best calibration curve for a given situation.

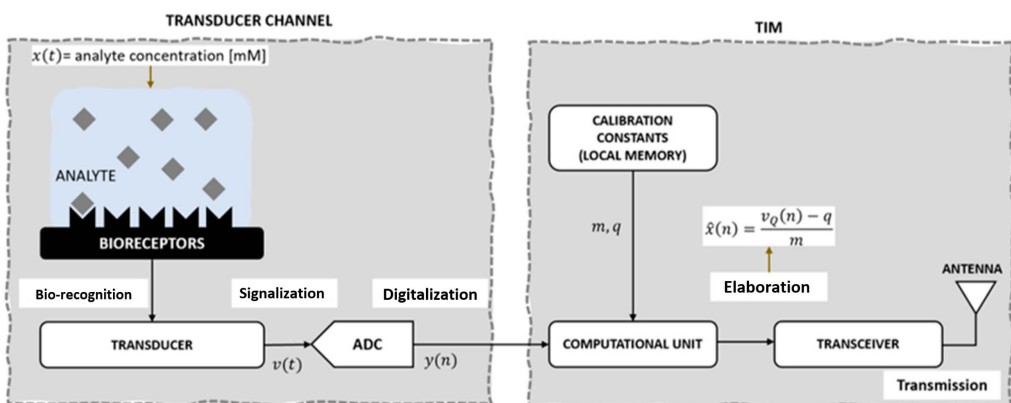

**Figure 5.** Functional diagram of a wearable biosensor.

Finally, the estimated quantity is transmitted to the final user through the wireless communication interface. Other solutions are also possible—for instance, we can directly transmit the output codeword $y(n)$ and calculate the estimated value of the analyte concentration in the remote app.

It is useful at this point to summarize the two basic kinds of transducers employed with biosensors. These parameters can eventually be tuned by the app designer to optimize the biosensor behavior—for instance, by increasing its sensitivity. As reported in [20], the kind of hardware and circuitry used in smart wearable biosensors depends on several factors, including the type of biosensor, the transducing principle (amperometric, impedimetric, or potentiometric), the data transmission method (e.g., RFID/NFC and Bluetooth) and the environment in which the device must operate. In general, the availability of portable, low-cost and miniaturized electronic circuits represents an important step toward a pervasive use of biosensors and, for this reason, their characterization assumed the same importance of that of the sensing device. In the following, among the various types of biosensors that could be integrated into smart wearable devices, only electrochemical sensors will be addressed. Compared to their main counterparts (mass-based and optical biosensors), electrochemical biosensors are easier to fabricate, miniaturize, and integrate on the same sensing substrate with customized readout circuits [21]. Further, recent advances in the area of printing technologies combined with advances in bio- and electrochemistry, nanostructures, solid-state and surface materials physics, integrated circuits, microfluidic and data processing enabled the possibility to address a whole new generation of smart electrochemical biosensors [2,22]. The most common classification of electrochemical biosensors divides them into impedimetric, amperometric and potentiometric biosensors. While for impedimetric biosensors, the measurement is commonly performed with a circuit that resembles standard impedance analyzers, the other two categories of electrochemical biosensors require the use of a proper transducer, called a *potentiostat*, to guarantee the proper conditioning of the system and to measure the electric signal affected by the analyte concentration [23]. In general, they are mainly distinguished by the signal provided—a current for amperometric biosensors and a voltage for potentiometric biosensors.

The logical structure of the transducer to convert the output of an amperometric biosensor in a voltage signal $v(t)$ is reported in Figure 6a. This circuit is based on a three-electrode layout characterized by a *working electrode* (WE) on which the chemical reaction takes place, a *reference electrode* (RE) that provides a constant voltage potential that does not depend on the kind of electrolyte solution—no current flows through this electrode—and finally a *counter electrode* (CE) to measure the current which flows through the electrochemical cell. This transducer is composed of two elements: a *control circuit* and a

*current-to-voltage (I/V) converter.* The control circuit, based on an operational amplifier with a negative feedback loop, regulates the voltage potential of the electrochemical cell to achieve a constant value. The I/V converter is employed to measure very low currents, reducing the effects due to variations in output impedance. The parameters that describe the behavior of this transducer—*sensitivity*, which is measured in Ω and the *bias potential*—can eventually be modified.

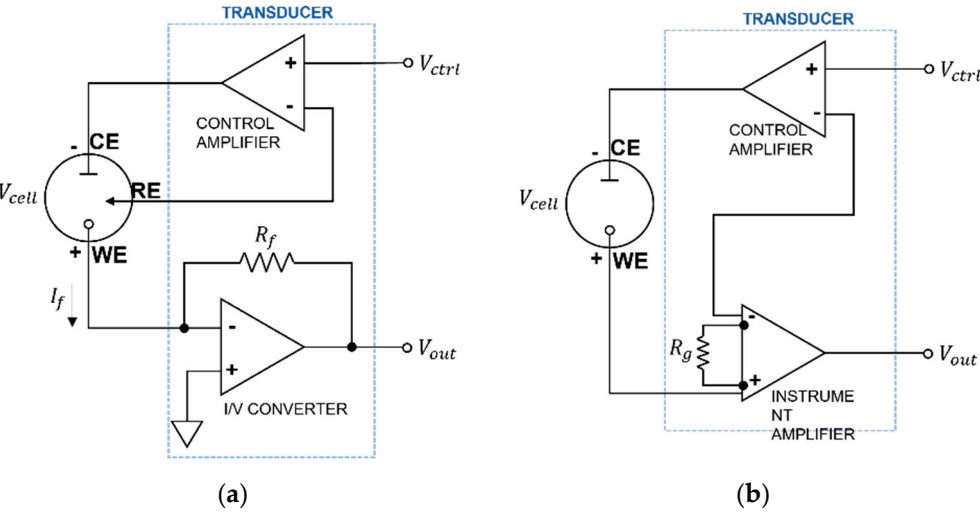

(**a**)                                                           (**b**)

**Figure 6.** Left (**a**): a functional diagram of the transducer for amperometric biosensors. Right (**b**): a functional diagram for potentiometric biosensors.

The transducer for a potentiometric biosensor is based on a two-electrode layout, as illustrated in Figure 6b. The conditioning circuit is composed of two elements: a *control circuit* and an *instrument amplifier*. The control circuit is composed of an operational amplifier with a negative feedback loop to keep the bias potential of the electrochemical cell stable. The instrument amplifier amplifies the differential input voltage between the electrodes WE and RE by guaranteeing, at the same time, a very high input impendence. The parameters that should be specified in this case are the *bias potential* and the *gain* of the instrument amplifier, which, in this case, is an a-dimensional number.

**4. TEDS for Smart Biosensors**

There are some important parameters that must be specified to describe the behavior and performance of a smart biosensor. For a generic smart sensor, the Dot 0 standard defines a set of transducer electronic data sheets (TEDSs). A TEDS is a digital file containing blocks of information and usually stored in the non-volatile memory within a TIM. The Dot 0 standard enables also storing the TEDS in other places of the user system when the former solution is not practical or is excessively resource consuming. In this case, these electronic documents are referred to as *virtual TEDS*. As a rule, a TEDS is not changed once the manufacturer or the system designer has established its contents.

All TEDSs have the same structure, as illustrated in Figure 7. The first field *length* (4 octets) specifies the total number of octets in the data block and checksum fields. The last block *checksum* is useful to verify the correctness of the TEDS files. The second field, *data block*, contains the information organized on the basis of the type/length/value (TLV) data structure: *type* identifies the field in the specific TEDS, *length* provides the number of octets in the value field and finally *value* contains the TEDS information.

Type 02 and type 03 have the same meaning for all the TEDSs. In particular, type 03 corresponds to the TEDS identifier, the corresponding length is equal to 4 octets and the value field is composed of 4 parameters: *family* identifies the member of the IEEE 21451 family of standard (e.g., value = 0 for the Dot 0), *class* identifies the kind of TEDS and it represents the TEDS access code (e.g., 1 for Meta TEDS, 3 for transducer channel

TEDS, and 5 for Calibration TEDS), *version* is the standard release (e.g., 1 corresponds to the original Dot 0 release), and *tuple length* specifies the number of octets in the length field of the TLV data structures for a given TEDS (usually this value is equal to 1).

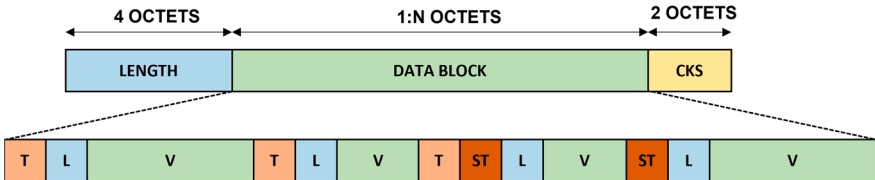

**Figure 7.** TEDS file format: T = type, ST = subtype, L = length, V = value, and CKS = checksum.

There are some mandatory TEDSs specified in the *Dot 0* standard, while others are optional. Two of the most important mandatory TEDSs are: *transducer channel* (TCh) and *Meta* TEDS. The former provides detailed information about a specific transducer channel. The second describes how the physical sensors are organized within the same TIM and eventually it specifies the dependencies among these sensors. In the following, we analyze the parameters that are specified in the current TEDS to understand the changes required to extend these standards also to smart biosensors.

### 4.1. Meta TEDS

The Meta TEDS is associated with each TIM and it provides common information for all the transducer channels, the number of sensors and actuators and how they are logically grouped. Table 1 is an example of Meta TEDS that describes the sensing system illustrated in Figure 4. As it can be noted, sEMG sensors can be organized in different proxy channels, a specific TCh identification number is associated with each proxy and it can be used to address a specific group.

**Table 1.** Example of data block for the Meta TEDS related to sensing systems illustrated in Figure 4. Data are represented as hexadecimal numbers; xx represents a generic hexadecimal value.

| Type | ST | Length | Value | Description |
|------|----|--------|-------|-------------|
| 03 | | 04 | 00-01-01-01 | TEDS identifier: family = 0, class = 1, vers = 1, t-length = 1 |
| 04 | | 0A | xx-xx-xx-xx-xx-xx-xx-xx-xx-xx | UUID universal unique id (see the Dot 0 standard) |
| 0D | | 02 | 00-09 | Specifies the number of TChs in the TIM: 9 |
| 12 | | - | - | Identifies the beginning of proxy data structures |
| - | 15 | 08 | 00-01-00-02-00-03-00-04 | TCh member list: sEMG CH 1-2-3-4 |
| - | 16 | 02 | 00-0A | TCh proxy id: 0x000A |
| - | 17 | 01 | 01 | Dataset organization: interleave |
| - | 15 | 08 | 00-05-00-06-00-07-00-08 | TCh member list: sEMG CH 5-6-7-8 |
| - | 16 | 02 | 00-0B | TCh proxy id: 0x000A |
| - | 17 | 01 | 01 | Dataset organization: interleave |
| - | 15 | 08 | 00-01-00-08 | TCh member list: sEMG CH 1-8 |
| - | 16 | 02 | 00-0C | TCh proxy id: 0x000C |
| - | 17 | 01 | 01 | Dataset organization: interleave |
| - | 15 | 08 | 00-02-00-07 | TCh member list: sEMG CH 2-7 |
| - | 16 | 02 | 00-0D | TCh proxy id: 0x000D |
| - | 17 | 01 | 01 | Dataset organization: interleave |
| - | 15 | 08 | 00-03-00-06 | TCh member list: sEMG CH 3-6 |
| - | 16 | 02 | 00-0E | TCh proxy id: 0x000E |
| - | 17 | 01 | 01 | Dataset organization: interleave |
| - | 15 | 08 | 00-04-00-05 | TCh member list: sEMG CH 4-5 |
| - | 16 | 02 | 00-0F | TCh proxy id: 0x000F |
| - | 17 | 01 | 01 | Dataset organization: interleave |

A proxy is an artificial construct to combine the outputs of multiple sensors into a single structure. A unique TCh number is associated with each proxy structure, which is

employed to simultaneously read all the samples acquired from the related channels. For instance, the proxy 0x000A enables simultaneously collecting data from sEMG channels CH1-2-3 and 4. Conversely, the proxy 0x000E can be employed to acquire data from only two sEMG channels: CH3 and 6. In this way, an app designer can easily select the group of sensors by simply reading from the related proxy channel. It is important to note that these groups are specified directly by the manufacturer. Samples acquired from the transducer channels with the same proxy structure are stored in a single dataset and they can be organized into two possible methods: *block* or *interleave*. Let us assume that $x^j(n)$ corresponds to the sample acquired by the $j$-th sensor in the time instant $n$, where $j = 1, \ldots, J$ represents the sensor id number and $n = 0, \ldots, N-1$ the time instant in the observed time window, where $N$ is the number of acquired samples. With a block structure, data are organized as follows: $\left[x^1(0), \cdots, x^1(N), \cdots, x^J(0), \cdots, x^J(N)\right]$, while data with an interleave structure are organized by interleaving the samples acquired from the different sensors: $\left[x^1(0), \cdots, x^J(0), \cdots, x^1(N), \cdots, x^J(N)\right]$.

Furthermore, Meta TEDS enables specifying other dependencies among sensors and other devices. This gives the possibility of defining *control groups*—for instance, where some transducers are used to control the operation of others. For example, a Meta TEDS can be employed to describe how to set the parameters of the potentiostat connected to the biosensor. A control group is a hierarchical structure composed of the subfields *group type* and *member list*. The standard already defined nine different group types, even though other groups can be defined by the manufacturer. For instance, type 1 is used to identify the embedded actuator used to setup an event sensor and its elements are reported in Table 2 for a better understanding. Type 2 identifies the embedded actuators used to set the high-pass filter, the low-pass filter and the scale factor associated with a given sensor. Type 3 is used to identify an embedded actuator to set the ADC sampling period. Finally, the member list is an ordered array that specifies the list of the transducer channel numbers for the transducers that perform each function.

**Table 2.** Type 1 control group specified in the Meta TEDS.

| Enumeration | Member List Order | Function |
|---|---|---|
| | 1 | Analog event transducer channel |
| 1 | 2 | Analog input sensor t.ch. This sensor measures the input value for the same input as member 1, thus providing the state |
| | 3 | Upper threshold embedded actuator transducer channel |
| | 4 | Hysteresis embedded actuator transducer channel |

This control group is composed of an analog sensor, an analog event sensor and two actuators.

Currently, no control group proposed by the standard is suitable for controlling the parameters of the transducers previously introduced for amperometric and potentiometric biosensors and a new type of control group should be introduced, as suggested in Table 3. In this case, a unique structure is proposed to represent every kind of transducer. However, since the member list is an ordered array, no elements may be omitted and when a transducer channel is not required, the corresponding number shall be set to zero. This structure is useful when transducer parameters can be tuned by the user.

### 4.2. Calibration TEDS as Defined by the Dot 0 Standard

In this section, starting from the Calibration TEDS as already defined by the Dot 0 standard, we provide a solution that enables using this TEDS to provide information about the calibration curve to an app designer. In the next section, an ad hoc solution to address the problem of calibration in biosensor is presented.

**Table 3.** Proposed control group for representing a transducer.

| Enumeration | Member List Order | Function |
|---|---|---|
| x | 1 | Analog sensor transducer channel (biosensor). |
| | 2 | Embedded actuator transducer channel to control the bias potential. |
| | 3 | Embedded actuator transducer channel to control the sensitivity of the I/V converter in the potentiostat for amperometric biosensors. |
| | 4 | Embedded actuator transducer channel to control the gain of a programmable gain amplifier. |

This control group is composed of an analog sensor and four different actuators.

The concentration of a biochemical substance can be indirectly determined by considering a given set of reference values that are measured at the biosensor output when a known concentration of the analyte is applied in input. These values can be represented by a calibration graph: input quantities (*x*-axis) correspond to the known concentrations of the analyte, expressed in molar (M) unit or its submultiples, where in SI measurement units 1 mM = 1 mmol/L = mol × m$^{-3}$. The output quantity depends on the kind of biosensor; for instance, amperometric biosensors provide a current (mA, μA or nA) and potentiometric biosensors provide a voltage (mV or μV).

Figure 8 provides a calibration graph example for a lactate biosensor, where the input quantity corresponds to the lactate concentration and the output is a current. For a generic sensor, the set of points reported in the calibration graph can be represented in a more concise way through a mathematical formula that approximates the behavior of the sensor over a given input interval. This formula can be subsequently employed to estimate the physical quantity applied in input. For a very simple linear sensor, the formula corresponds to the straight-line equation:

$$y = mx + q \tag{1}$$

where *x* is the quantity in input and *y* the corresponding output. *m* and *q* are, respectively, the sensitivity of the sensor and the corresponding offset value, i.e., the value observed in output from the sensor when a nil value is applied in input. For a generic sensor, these parameters are provided in the *Calibration TEDS* [12].

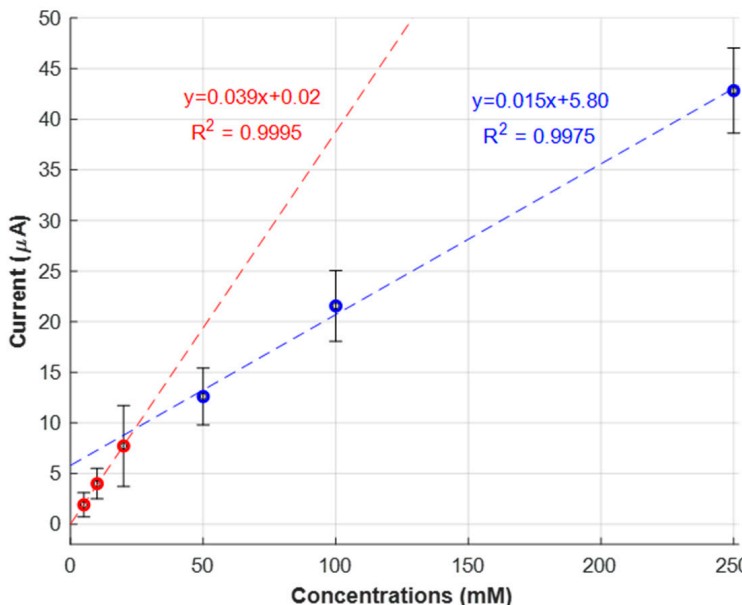

**Figure 8.** Example of the calibration curve for a laboratory-made printed amperometric biosensor for lactate quantification.

It is important to remark that this TEDS is considered optional by the *Dot 0* standard, since smart sensors directly provide the estimated quantity $\hat{x}$, calculated within the TIM by using the inverse formula of (1). For a biosensor, the Calibration TEDS should be mandatory since it contains information that an app designer could exploit to improve the accuracy of the estimated value in different contexts. The calibration curve for a biosensor can be approximated, as suggested by Figure 8, by different interpolating functions depending on the considered input range. It can be observed that, for a linear sensor, the number of segments is only one, which means that they can be described by a single equation over the whole input range. In this case, the input–output relationship is completely specified by the two parameters $m$ and $q$, expressed in SI units.

One or more segments need to be defined instead when the calibration graph is approximated by a piecewise function, as for a biosensor. This can be easily achieved by considering some of the parameters already defined in the Calibration TEDS, as already explained. The parameter *array of low boundary limits* is a one-dimensional array $[B_1, \cdots, B_s, \cdots, B_K]$ that contains the lower boundary for each segment, in ascending numerical order: $B_1 < B_2 < \cdots$, where $K$ represents the total number of segments. The first element of the array $B_1$ corresponds to the lowest value in input. In general, given the input value $x$, to identify the segment at which it belongs, it is sufficient to check when it satisfies the following equation: $x \in S \rightarrow B_s \leq x < B_{s+1}$. Offset values are then provided by the TEDS parameter *segment offset values*, which is a one-dimensional array containing one offset value for each segment: $[q_1, \cdots, q_s, \cdots, q_K]$. Sensitivity values are specified instead in the TEDS entry defined *set of coefficients*. Since polynomial functions can also be employed to interpolate the calibration graph, this parameter is a matrix; however, for a linear sensor, it corresponds to a one-dimensional array: $[m_1, \cdots, m_s, \cdots, m_K]$, where each element represents the sensitivity for a given segment. For instance, considering the lactate biosensor of Figure 3, the calibration curve can be described assuming piecewise linear behavior for this sensor. The result: $B = [0, 30]$ mM, $q = [0.02, 5.80]$ µA and $m = [0.39, 0.15]$ µA/mM.

The Calibration TEDS contains other useful information that enable a periodic sensor recalibration. A periodic calibration enables achieving a more accurate estimation of the concentration of a given substance by periodically adjusting the calibration curve. This operation is quite different compared to the initial calibration procedure, during which the calibration graph is obtained as previously introduced and it aims essentially to correct the offset value. In fact, while the sensitivity is assumed to be stable for the whole period guaranteed by the manufacturer, the offset needs to be adjusted by the user with a simple offset adjustment procedure that can be performed locally. This procedure consists of applying in input a black control, which corresponds to a nil input $x = 0$; thus, estimating the actual offset: $\hat{q} = y(x = 0)$, where $y$ corresponds to the output value. Recalibration is usually performed at periodic time intervals depending on the calibration period. The calibration period is determined by the manufacturer during the shelf stability test and should be specified in the Calibration TEDS. Table 4 reports the related parameters that can be found in the Calibration TEDS: the *last calibration date* and the *calibration interval*. The former specifies the last calibration date. The latter provides the time interval, in seconds, during which the sensor can operate within the limits for the operational uncertainty and therefore it does not need to be recalibrated. In the current version of Calibration TEDS, a single value is reported. However, for a biosensor, it could be useful to consider two different parameters, one for the stability that determines the lifetime of a biosensor and a second for the offset that can be used instead to decide when an offset recalibration is needed. A calibration period equal to a nil value indicates that an offset recalibration is always needed before using the biosensor for the first time, especially for disposable biosensors.

**Table 4.** Calibration TEDS structure for a generic sensor.

| Name | Description | Type |
|---|---|---|
| Last calibration | It specifies the last calibration date. | Time Instance [1] |
| Calibration period | It specifies the calibration interval and depends on the stability of the sensor. | Time Duration [2] |

[1] Time Instance: structured data type to represent a time value and not a time duration; it is composed of two fields: nanoseconds and seconds, both represented as Uint32. The epoch is defined as starting at midnight 1 January 1970. [2] Time Duration: this structured data type is a subclass of time representation which is used to specify a time interval rather than a time value; it is composed of two fields: nanoseconds and seconds, both represented as Uint32.

### 4.3. Manufacturer-Defined Biosensor Calibration TEDS

As highlighted by analyzing the calibration graph, for each input concentration employed as a reference value, the corresponding biosensor can assume different output values within a given range. As illustrated in Figure 3, the output values can be parsimoniously described by considering at least two values, e.g., (*mean*, *standard deviation*) for symmetric distributions or more values for asymmetric ones—for instance (*mean*, *min*, *max*) or (*mean*, *standard deviation*, *min*, *max*), etc. However, the Calibration TEDS defined in the Dot 0 standard does not provide a data structure where these values can be stored. For this reason, it is very useful to consider the manufacturer-defined TEDS. This electronic document can be in any format required by the manufacturer to enable a given application. This TEDS content and structure are defined by the manufacturer. The access code for this kind of TEDS can be any number between 128 and 255. In the case of biosensors, it is useful to define a specific electronic data sheet in which all the information obtained during the laboratory calibration can be saved. This information, however, depends on the kind of calibration performed in the laboratory to obtain the corresponding graph. In the following, we briefly describe these two kinds of calibration methods and then we propose a suitable TEDS for storing the related calibration data.

These two methods are: *single concentration chronoamperometry or potentiometry* and *standard addition-based chronoamperometry or potentiometry*. The first method considers a given number of identical sensors, one for each concentration that must be reported in the calibration graph. In the example of Figure 9, the calibration is performed by considering 6 different biosensors of the same kind. A different concentration of the targeted analyte is applied in input of each sensor, e.g., 0 mM (blank control—blk), 5, 10, 20, 50, 100 and 250 mM. The output signal (the current for amperometric biosensors and the voltage for potentiometric biosensors) is measured through a single acquisition referred to as chronoamperometry for amperometric sensors and potentiometry for potentiometric sensors. Finally, the corresponding output value is measured at the end of the initial transitory, when the output has reached the stationary condition and it remains within a given convergence region, as shown in the zoom-in view in Figure 9. Statistical analysis of the output signal can be performed by observing the output value over a given window.

The second method for performing the initial biosensor calibration is based on a single chronoamperometry or potentiometry during which standard additions of known volumes of solution are performed to obtain given concentration steps. In this case, the calibration can be performed by considering a single sensor at which the substance of interest is added during successive time instants. Before proceeding with the successive addition, we must wait until the biosensor has reached a stationary state to measure the output value in correspondence of a given input concentration. This kind of calibration requires a long chronoamperometry or potentiometry to acquire all the biosensor output values [24]. An example is given in Figure 10 for a lactate sensor that can be easily extended to other classes of electrochemical biosensors. In this example, four different input substance concentrations have been considered, 0, 40, 60 and 80 mM; these values and the corresponding addition time instants are reported in the top-left graph, which provides the stimulus profile applied for calibrating the sensor. On the right we can see the final calibration curve. Reproducibility was evaluated by repeating the same calibration procedure for four different biosensors.

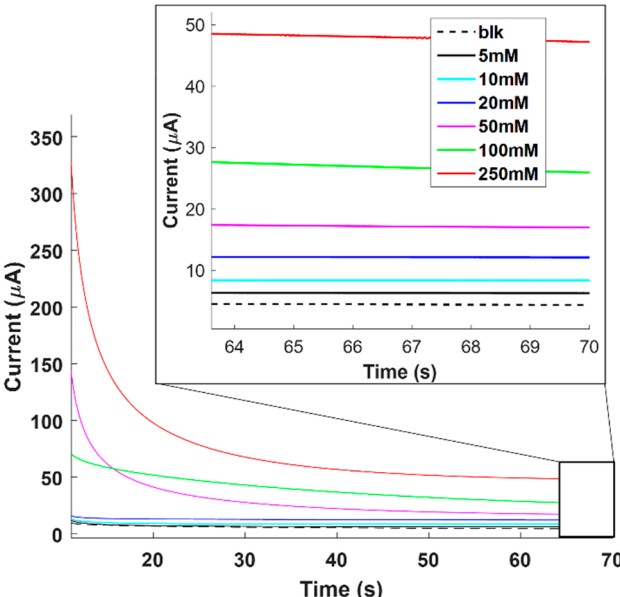

**Figure 9.** Calibration example based on multiple sensors for each concentration. The experimental values have been obtained for a flexible lactate biosensor. The corresponding output current is measured after the system has reached the steady state, that is after approximately 65–70 s.

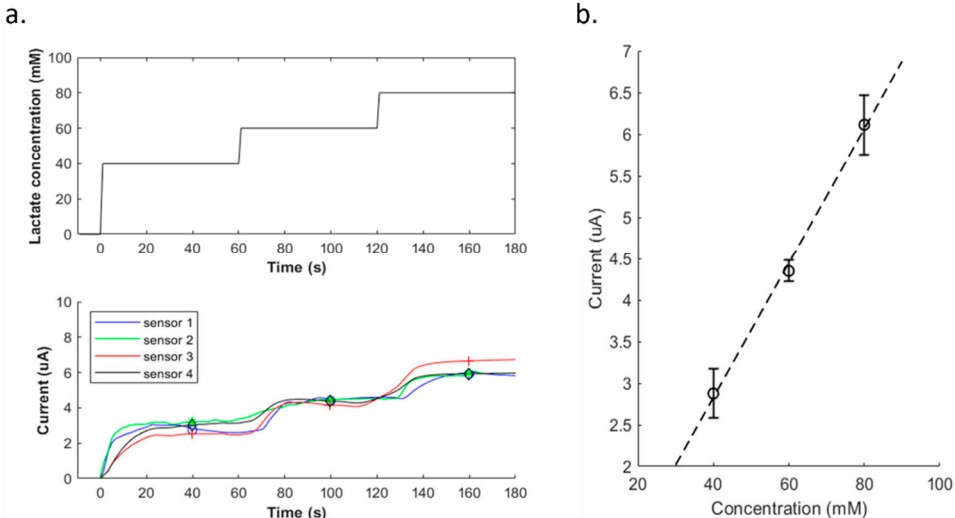

**Figure 10.** Calibration example by using a standard addition method. The experimental values have been obtained for a flexible lactate biosensor. In subfigure (**a**) the concentration changes applied are presented in the top graph and the measured output current in the bottom-left graph with the points (diamonds) at which the steady-state current value is measured. In subfigure (**b**) the corresponding calibration curve obtained is reported.

Table 5 provides an example of a possible manufacturer-defined Calibration TEDS that we wish to propose based on our expertise in the field of biosensors. This document provides an exhaustive representation of all the information that is reported in a generic biosensor calibration graph, independently from the calibration method. Without lacking in generality, in this example, the access code 80 is employed to identify this kind of TEDS. Two parameters can be considered mandatory: the first (type 10) provides the number of points reported in the calibration graph, which also corresponds to the number of effective concentrations for which the biosensor was tested. This information is very important because it provides the final user with more insights about the quality of the calibration procedure.

**Table 5.** Example of manufacturer-defined Calibration TEDS for a biosensor.

| Type | ST | Length | Value | Description |
|------|-----|--------|-------|-------------|
| 03 | - | 04 | 00-80-01-01 | TEDS id |
| 10 | - | 01 | 03 | Number of points in the calibration graph |
| 11 | - | - | - | Point list |
| - | 20 | 02 | xx-xx | Input reference value |
| - | 21 | 02 | xx-xx | Mean output value |
| - | 22 | 02 | xx-xx | Minimum output value |
| - | 23 | 02 | xx-xx | Maximum output value |
| - | 24 | 02 | xx-xx | Standard deviation for the output values |
| 12 | - | 0B | xx-xx-xx-xx-xx-xx-xx-xx-xx-xx-xx | SI unit for input values |
| 13 | - | 0B | xx-xx-xx-xx-xx-xx-xx-xx-xx-xx-xx | SI unit for output quantities |
| 14 | - | - | - | Calibration method related information |
| - | 30 | 01 | xx | Method kind: chronoamperometry (01), potentiometry (02), standard additions (03) |
| - | 31 | 01 | xx | Number of sensors (only for methods 01 or 02) |

The second element (type 11) is structured data and it is composed of five subfields: the input reference value, the mean, the minimum and maximum output values, and the standard deviation for the output values. This data structure is specified for every point in the calibration graph. Finally, elements associated with type 12 and 13 specify the SI measurement unit for both input quantity and output quantity.

Optional parameters can be specified when the manufacturer wants to show the quality of the calibration procedure performed in the laboratory. An example is provided by type 14, which corresponds to a data structure for specifying the calibration method and reporting information about the calibration protocol employed.

*4.4. Transducer Channel TEDS*

This electronic document contains all the information for complete biosensor characterization, the electronic circuit and the related process for data acquisition, including parameters related to the analog-to-digital converter. As reported in the Dot 0 standard, the TCh TEDS is divided into two macro blocks, the former reporting the transducer channel-related information and the latter the data-converter-related information. For the second block, there are no substantial differences between a biosensor and a generic sensor. The first block should instead be modified to also report important parameters for the characterization of a biosensor. Table 6 reflects the proposed variations compared to the current TCh.

**Table 6.** Transducer channel TEDS structure for a generic sensor (white) integrated with information that should be provided for a biosensor (gray).

| Type | ST | Length | Value | Description |
|------|-----|--------|-------|-------------|
| 03 | - | 04 | 00-03-01-01 | TEDS id |
| 0B | - | 01 | 00 | Kind of a transducer channel: sensor = 0, actuator = 1, event sensor = 2 |
| 0C | - | 0B | xx-xx-xx-xx-xx-xx-xx-xx-xx-xx-xx | Measurement unit for the measured data (sensor) |
| 0D | - | 04 | xx-xx-xx-xx | Lower range limit in input for the sensor (in SI unit) |
| 0E | - | 04 | xx-xx-xx-xx | Higher range limit in input of the sensor (in SI unit) |
|  |  | 04 | xx-xx-xx-xx | Limit of detection |
|  |  | 04 | xx-xx-xx-xx | Limit of quantification |
|  |  | 01 | xx | Selectivity, in percentage, for each interference substance. |
| 0F | - | 04 | xx-xx-xx-xx | Worst-case measurement uncertainty (in SI) introduced by the sensor in the worst case, by considering any quantity that may vary during the measurement process |
|  |  | 04 | xx-xx-xx-xx | Repeatability: intercomponent variability due to the fabrication process |

**Table 6.** *Cont.*

| Type | ST | Length | Value | Description |
|---|---|---|---|---|
| | | 04 | xx-xx-xx-xx | Reproducibility: precision for a calibrated sensor |
| | | 04 | xx-xx-xx-xx | Sensibility: ability of a sensor to keep its characteristic unaltered during the time |
| | | 04 | xx-xx-xx-xx | Response time: parameter to describe the dynamic behavior of a sensor |
| 12 | - | - | | Information concerning the sampling and quantization processes |
| - | 28 | 01 | 00 | Data model for the dataset: N-octet unsigned integer =0 |
| - | 29 | 01 | 02 | Data model length: # octets to hold 12 bits = 2 |
| - | 2A | 02 | 0C | Significant bits in the data model: 12 bits |
| 13 | | | | Information concerning the dataset structure |
| - | 2B | 02 | 00 | Max number of repetitions L that can be acquired during a single acquisition after the first sample |
| - | 2C | 04 | - | Series origin: it represents the value of the independent variable associated with the first sample in the dataset. It is optional if L = 0. |
| - | 2D | 04 | - | Series increment: it represents the minimum spacing between values of the independent variable associated with the repetitions stored in the dataset (e.g., sampling period in the case of a uniform time sampling). Omitted for L = 0 |
| - | 2E | 0B | xx-xx-xx-xx-xx-xx-xx-xx-xx-xx-xx | Series unit: SI unit for series origin and increment |
| - | 2F | 02 | 00 | Maximum pre-trigger samples: only in the case of an acquisition operating in free-running with pre-trigger mode. It represents the max number of samples that can be acquired before a trigger event |
| 1F | - | - | | Sampling |
| - | 30 | 01 | 04 | Sampling modes: it describes the sampling modes that are supported by the transducer channel (0 = trigger-initiated mode, 1 = free-running without pre-trigger, 2 = free-running with pre-trigger, 3 = continuous mode, 4 = immediate mode) |
| - | 31 | 01 | 04 | Default sampling mode |
| 20 | - | 01 | 01 | Data transmission mode: it specifies how data are transmitted by the sensor (1 = command-based, 2 = when buffer full, 3 = at fixed intervals) |

UInt8: unsigned integer on 8 bits. UNITS is structured data composed of 11 fields, each one represented by a UInt8. A total of 10 fields correspond to the exponent for the corresponding SI fundamental units. Another field is provided to specify the multiplicative factor of multiples or submultiples as a power of 10. Float32: single-precision real numbers on 32 bits.

For biosensors, the measured quantity corresponds to a concentration, expressed in molars fractions (usually µM or mM) and corresponding to the series unit parameter: [0 0 0 0 −3 0 0 0 0 1 0]. Each element in this array represents the exponent of the corresponding SI fundamental units: radiant, steradian, meter, kilogram, second, ampere, kelvin, mole, and candelas [14].

The lower and higher value limits for the input range depend on the minimum and maximum values that can be measured with the sensor; for instance, focusing on the example provided targeting lactate in sweat, considering that its content in undiluted sweat usually ranges from 0 to 100 mM [25], we can assume a minimum value equal to 0 mM (negative numbers are meaningless in this case) and a maximum value of 100 mM. For a generic sensor, the TCh TEDS reports the worst-case uncertainty. For a biosensor, two important parameters should be provided, i.e., *limit of detection* (LOD) and *limit of quantification* (LOQ). The former provides an indication about the capability of the biosensor to detect the presence of a target analyte and the second instead also quantifies the amount of analyte applied in input. These parameters take into consideration both the slope of the calibration curve (sensitivity) and the intrinsic variability of the sensor without the target analyte applied in input—*blank control*. In particular, the LOD is calculated using the so-called "3σ rule" [26]. This procedure consists of evaluating the standard deviation, on

the output of the sensor during repeated measurements performed on a blank sample. This value is then multiplied by the numerical factor 3 and finally the corresponding analyte concentration is obtained by multiplying this value by the sensitivity. The LOQ is computed instead following the same protocol but using the "10σ rule" [15] or otherwise 3 times the LOD [27].

Other parameters that are usually considered to characterize the behavior of a biosensor are referred to as interference substances and the corresponding selectivity, repeatability and reproducibility, since in this kind of sensor they usually assume very large values, stability and the response time of the sensor [28], as illustrated in Table 6 in gray. These parameters are summarized in the following for completeness.

*Selectivity* characterizes bioreceptors and represents the capability of these components to specifically detect a given analyte in a sample containing a mixture of other analytes and contaminants. A specific biosensor ideally responds only to a single analyte, recognizing no other. This behavior is mainly approached by antibodies, aptamers, and enzymatic lock-and-key or bioconjugation pairs. The selectivity of a specific biosensor is expressed with an a-dimensional scalar number, usually a percentage [29]. The output of an ideal biosensor varies only when there is variation in the concentration of the target analyte; conversely, the output remains constant. In real biosensors, the output is affected, albeit to a lesser extent, by variations in the concentrations of other analytes or interferents. The selectivity quantifies the variation in output due to a different analyte or interferent. For example, a selectivity of 10% indicates that, when an interferent is added, the output exhibits a variation of 10% compared to the ideal behavior.

The ideal behavior of highly specific biosensors is very often not achievable, because of the high similarity between the conformation or the reactivity of different analytes. A possible solution to this problem was discussed in the literature, and it is mainly based on array-based biosensors, composed of many sensing elements, where each element interacts somehow with the analytes of interest, thus creating a fingerprint for the measure. Array-based biosensors are single-input multiple-output (SIMO) systems, where the output is multidimensional data [30]. An important advantage of the array-based biosensor approach is that these devices can be easily adopted for the analysis of different analytes without or with very few changes. For example, to recognize new analytes, it is sufficient to update the pattern recognition algorithm by retraining the array on known samples. Examples of these synthetic arrays have been widely used in the literature for the detection of small molecules, explosives, drugs, and other biological samples such as proteins, cells, and bacteria.

For TCh TEDS, to express the selectivity of array-based biosensors, we need an ordered vector of numerical values. Each vector element will represent the selectivity compared to the corresponding analyte and the sequence of analytes must be known to the user and also specified in the electronic document.

The other two key parameters for the characterization of the performance of a biosensor are *repeatability* and *reproducibility*. *Repeatability* describes the intercomponent variability (tolerance) due to the fabrication process and depends on the biofunctionalization and the chemical equilibrium at the interface. Repeatability refers to the ability of a given sensor to respond in the same way under identical input stimuli, working conditions, and measurement setup. Repeatability corresponds to a relative standard deviation and therefore its measurement unit is an a-dimensional number, typically a percentage. Specific protocols for evaluating repeatability are discussed in the literature [31,32], which enable restoring exactly the same initial condition before starting each measurement used to evaluate this parameter.

*Reproducibility* is the capability of similar sensors to respond in the same way when subjected to the same input stimulus, under the same working conditions. For example, the reproducibility of a lactate sensor depends on the reproducibility of the intrinsic electrical properties of the electrodes and the amount of enzyme and the reproducibility of its deposition conditions. The main disadvantage of using poorly reproducible sensors is that they need to be frequently re-calibrated. As already discussed, calibration is a necessary

procedure to compensate for sensor-to-sensor fabrication variations, thus improving the accuracy of a sensor. Periodic recalibrations must be considered to reduce sensor drift, and they are strictly related to the stability of a sensor. Calibration is among the main issues that limit the use of biosensors and is currently a hot interest of recent literature on the topic [33].

*Stability* represents the degree of susceptibility to environmental disturbances and other factors that could take place in and/or around the biosensing system. The output of an unstable biosensor typically presents a drift that affects the quality of the measurement information [28]. This parameter is usually expressed as the maximum variation in the output of the sensor after a predefined time interval. Similarly, stability could also be provided as the first derivative of the output during the time; for instance, the measurement unit for amperometric biosensors is mA/h, while for potentiometric biosensors is mV/h when expressed compared to the output. Otherwise, if the variation is provided by considering the input, the measurement unit is mM/h.

It is possible to distinguish between two distinct kinds of stability: *shelf stability* and *operational stability*. The former is usually evaluated over longer time periods, typically months, through a quite simple periodic measurement repeated after a long time interval from the initial calibration. The purpose of this parameter is to estimate the possible degradation and the need for a possible recalibration before using the biosensor for the first time [34]. The latter is evaluated over short periods of time, typically hours, through a continuous measurement to estimate the changes that might take place during a continuous operation.

Finally, the *response time* is defined as the time that the output of a sensor needs to settle at a final value within a tolerance band when the corresponding input quantity is changed. The corresponding measurement unit is seconds. This parameter is useful for characterizing the dynamic properties of a biosensor and should therefore be carefully considered in the development of applications based on biosensors [35].

## 5. Conclusions

In this paper, we have provided an exhaustive analysis of the main characteristics of a biosensor that should be provided to a final user to guarantee proper use of that sensor. Following the guidelines already available to standardize other kinds of sensors, we have proposed a feasible way to make biosensors smart, thus providing stand-alone and modular devices that could be easily integrated into wearable devices. A significant advantage comes from the flexibility endowed to the wearable biosensors by the possibility of connecting different sensing devices according to a plug-and-play scheme.

The applicability of the ISO/IEC/IEEE 21451 family of standards is by no means restricted to traditional transducer networks. In fact, it is particularly stimulating to think of a smart e-health monitoring system as a new and challenging way of defining the wireless sensor network concept. Capitalizing on the experience of the ISO/IEC/IEEE 21451 standard can provide benefits in terms of efficient, robust, and effective implementations. Furthermore, the IEEE 21451 architecture is based on open standards, allowing developers to seamlessly add additional required features, as proposed in this document, to consider the peculiarities of biosensors. This feature enables hardware from different vendors to co-exist peacefully, improving the services offered to the end users.

Another notable strength of the proposed approach, based on the ISO/IEC/IEEE 21451 standard, is its independence from specific communication technologies and the very extensive use of electronic data sheets—TEDS—for exhaustively describing the parameters that characterize the behavior and performance of a biosensor. This would make an ISO/IEC/IEEE 21451-compliant monitoring system extensively documented and remotely configurable, supporting the concepts of robustness, adaptability, and ease of deployment, enabling personalized solutions. The standard ISO/IEC/IEEE 21451 was designed from its origin to support lightweight, low-cost implementations and can help to lower implemen-

tation costs, thus meeting a significant challenge for the acceptance and wide diffusion of the methods and technologies discussed in this work.

**Author Contributions:** G.G. and S.T. equally contributed to the conceptualization, investigation and writing of the original draft. All authors have read and agreed to the published version of the manuscript.

**Funding:** This research was funded by the Department of Information Engineering, University of Padua, Italy within the Networking Project: "SWEASE—Smart WEArable Sensors for E-Health applications".

**Institutional Review Board Statement:** Not applicable.

**Informed Consent Statement:** Not applicable.

**Conflicts of Interest:** The authors declare no conflict of interest.

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
