# Peer review of "Wearable Biosensor Standardization: How to Make Them Smarter"

_standards, doi:10.3390/standards2030025_

Round 1

Reviewer 1 Report

Dear Authors,

I have read your document in deepness.

By reading the title, it sounded quite an amazing paper to read/have/understand. However, while I started to read and to analyse your paper, I found quite a few issues. Most of them are stated on the attached file.

The main issue that I found is that there is not enough deepness for each of the topics, it sound as an academic report rather than a paper or review paper as I think it is.

The milestone of your paper are the protocols ISO/IEC/IEEE 21451; however, you never specify which is the core for such protocol or what is the main aim of it for bio-based devices.

Please read the attached file and properly address the suggestions. I reckon that your paper can be quite interesting but at the moment it is not. Please double check the entire document prior to re-submit it.

Author Response

We wish to thank the Reviewer for his/her precious suggestions. Please see the attachment for further details.

Reviewer 2 Report

Comments

In this article, the author attempts to establish the link between the ISO/IEC/IEEE 21451 standard and the development of wearable biosensors to analyze the further development direction of smart wearable biosensors.

  1. In this process, the author listed a series of characteristics of wearable sensors, including the main characteristics and parameters of biosensors, but did not provide enough logic in the overall structure of the article to support the relationship between ISO/IEC/IEEE 21451 and biosensors. And only a small amount of text is used to describe smart biosensors, so the important concept of smart biosensors is not presented in the full text.
  2. The figures in the article are also very rough. For example, the structure diagram shown in Figure 1 does not explain the main idea well. In these figures, there are also problems such as failure to explain the abbreviations and inconsistent text formats.
  3. I have also noticed that this article has the problem of too few citations for the existence of relevant documents.

Therefore I don’t recommend receiving this article.

Comments

In this article, the author attempts to establish the link between the ISO/IEC/IEEE 21451 standard and the development of wearable biosensors to analyze the further development direction of smart wearable biosensors.

  1. In this process, the author listed a series of characteristics of wearable sensors, including the main characteristics and parameters of biosensors, but did not provide enough logic in the overall structure of the article to support the relationship between ISO/IEC/IEEE 21451 and biosensors. And only a small amount of text is used to describe smart biosensors, so the important concept of smart biosensors is not presented in the full text.
  2. The figures in the article are also very rough. For example, the structure diagram shown in Figure 1 does not explain the main idea well. In these figures, there are also problems such as failure to explain the abbreviations and inconsistent text formats.
  3. I have also noticed that this article has the problem of too few citations for the existence of relevant documents.

Therefore I don’t recommend receiving this article.

Author Response

(The authors gave the same response as above.)

Reviewer 3 Report

I like to appreciate authors for their review ´Wearable Biosensors Standardization: How to make them smarter´ 

I find MS well written and all the aspects are well detailed and explained. I think it would be nice if you can add some examples of wearables which are available in the market or show some comparison. 

Author Response

We wish to thank the Reviewer for his/her encouraging opinion. In order to fix all the doubts and requested posed by the other Reviewers the paper has been completely re-written and ri-organized. In doing so, we have added a practical example of a real device, as suggested by the Reviewer. 

Round 2

Reviewer 1 Report

Dear Authors,

I am glad that you consider my suggestions for your re-submission. Attached are a few comments to your work. Along the text, there are common issues regarding the English punctuation.

As your mother tongue as a quite diverse, complex , exuberant and with loads of history in it, write in a language that is less complex that Italian sometimes it has some issues.

Your new submission is clean and punctual that the other one. Please take advantage of these suggestions and improve your paper.

Along the text there are quite a few opportunities to add bibliography to properly support your statements. A common syntax issue is presented along the text due to the aforementioned issue. A few issues in Figures 1 and 2 appear due to those were never called along the text and Fig. 3 needs to be completed, it seems that it requires further work. Whole images require far more quality at least 300 dpi. Also, the caption in whole of them requires further work.

Kind Regards

Author Response

We wish to thank the Reviewer for his/her precious suggestions and contributions in making this paper more interesting and with a higher technical quality. We have considered all the suggestions in the pdf file to fix errors and to improve the use of the English in our work.

Reviewer 2 Report

It can be accepted

Author Response

We wish to thak the Reviewer for his/her help.